# *Lactobacillus helveticus* Isolated from Raw Milk Improves Liver Function, Hepatic Steatosis, and Lipid Metabolism in Non-Alcoholic Fatty Liver Disease Mouse Model

**DOI:** 10.3390/microorganisms11102466

**Published:** 2023-09-30

**Authors:** Hyeonji Kim, Kippeum Lee, Ju-Yeon Kim, Jae-Jung Shim, Junghyun Lim, Joo-Yun Kim, Jung-Lyoul Lee

**Affiliations:** 1R&BD Center, hy Co., Ltd., 22, Giheungdanji-ro 24beon-gil, Giheung-gu, Yongin-si 17086, Republic of Korea; skyatk94@gmail.com (H.K.); joy4917@hanmail.net (K.L.); 10003016@hy.co.kr (J.-Y.K.); jjshim@hy.co.kr (J.-J.S.); 2Department of Pharmacy, School of Pharmacy, Jeonbuk National University, Jeonju 54896, Republic of Korea; jl1206@jbnu.ac.kr

**Keywords:** probiotics, non-alcoholic fatty liver disease, lactic acid bacteria, steatosis, β-oxidation

## Abstract

Here, we show that *Lactiplantibacillus plantarum* LP158 (LP158), *Lactobacillus helveticus* HY7804 (HY7804), and *Lacticaseibacillus paracasei* LPC226 (LPC226) isolated from raw milk alleviate non-alcoholic fatty acid disease (NAFLD) in a C57BL/6 mouse model. Lactic acid bacteria (LAB) were screened for their ability to inhibit fatty acid accumulation in palmitic acid (PA)-treated HepG2 cells, and three strains were selected based on the results. We also investigated hemolytic activity and antibiotic resistance of the three strains. LP158, HY7804, and LPC226 suppressed expression of mRNA encoding genes related to lipogenesis, and increased expression of genes related to β-oxidation, in a PA-induced HepG2 cell model. Moreover, when LP158, HY7804, and LPC226 were administered at 10^9^ CFU/kg/day for 8 weeks to mice with dietary-induced NAFLD, they all modulated blood biochemistry markers and reduced steatosis in liver tissue. Also, all three strains significantly reduced expression of mRNA encoding lipogenesis genes (*Fasn*, *Acaca*, and *Srebp-1c*) and inflammatory factors (*Tnfα* and *Ccl-2*) and fibrosis factors, and increased expression of a β-oxidation gene (*Acox1*) in the liver. In particular, HY7804 showed the strongest effects both in vitro and in vivo. Therefore, HY7804, LP158, and LPC226 can be proposed as potential supplements that can improve NAFLD through anti-steatosis, anti-inflammatory, and anti-fibrotic effects.

## 1. Introduction

Non-alcoholic fatty liver disease (NAFLD), a condition in which excess neutral fat accumulates in the liver, is the most common cause of chronic liver disease. The prevalence of NAFLD is estimated to be 20–30% worldwide, increasing to >70% in patients with obesity and diabetic risk factors [1,2]. Histologically, NAFLD encompasses a variety of liver diseases, ranging from simple steatosis to non-alcoholic steatohepatitis (NASH), liver fibrosis, cirrhosis, and liver carcinoma [3,4]. The earliest stage of NAFLD is simple steatosis, characterized by triglyceride levels that are 5% or more of liver weight [5]. The course of simple steatosis is almost spontaneous improvement, although in some cases it progresses to NASH [6,7].

The main causes of NAFLD are excessive calorie intake (mainly carbohydrates), obesity, and type 2 diabetes [8]. Usually, NAFLD is caused by excessive intake of carbohydrates or fats. Prolonged excessive intake results in hyperinsulinemia and hyperglycemia. Excessive intake of carbohydrates and fats increases decomposition of fatty acids in adipose tissue, which, in turn, increases the level of free fatty acids entering the liver [9,10,11]. In general, fatty acids introduced into the liver are used as an energy source via β-oxidation, or are converted into triglycerides and accumulate in the liver [12,13]. However, if hyperinsulinemia and hyperglycemia persist, fatty acid oxidation is suppressed and triglyceride synthesis is promoted [14,15]. Sterol regulatory element-binding protein-1c (SREBP-1c) is activated in the liver, and expression of genes that promote fatty acid synthesis is increased [16]. Furthermore, malonyl-CoA produced by increased fatty acid synthesis inhibits the action of carnitine palmitoyl transferase-1 (CPT-1), thereby reducing β-oxidation in the mitochondria [17]. Accumulation of fat in the liver causes liver inflammation, mediated mainly by release of cytokines and chemokines such as interlukin-1β (IL-1β), tumor necrosis factor-α (TNFα), macrophage marker (Cd68), and chemokine (Ccl2) [18,19]. Liver inflammation is an important precursor to liver fibrosis [20].

Probiotics, living microorganisms that are intended to have health benefits when administered in appropriate amounts, include strains of Lactobacillus, Bacillus, and Bifidobacterium, as well as some strains of Saccharomyces [21]. In particular, *Lactobacillus* species are associated with various benefits, including improved intestinal health, strengthening of the immune system, and antioxidant, anti-colitis, and anti-cancer properties. In addition, studies show that *Lactobacillus* species exert anti-obesity and anti-NAFLD effects. Recent studies show that *Lactobacillus* suppresses fat accumulation induced by a high-fat diet, as well as decreasing production of pro-inflammatory factors [22,23,24]. Being obese or overweight is the underlying cause of NAFLD in most patients, suggesting that probiotics may be an effective treatment [4,5,25].

This study aimed to investigate whether new potential probiotic strains *Lactiplantibacillus plantarum* LP158, *Lactobacillus helveticus* HY7804, and *Lacticaseibacillus paracasei* LPC226, isolated from raw milk, exert beneficial effects on NAFLD. These three species were screened from 27 LAB strains by evaluating their inhibitory effects on fat accumulation in lipid-accumulation-induced hepatocytes. We also examined the anti-lipogenesis and β-oxidation-activating effects of LP158, HY7804, and LPC226 in PA-treated HepG2 cells and in mice fed an NAFLD-promoting diet. Finally, histopathological and serological analyses were performed to examine the anti-inflammatory and anti-fibrotic effects of probiotic strains in diet-induced NAFLD mice.

## 2. Materials and Methods

### 2.1. Isolation, Culture, and Preparation of LAB Strains

Twenty-seven LAB strains were provided by the probiotics strain library of hy. Co., Ltd. All were isolated from raw fresh milk collected from domestic animals in the Yeongdong area of Korea. Raw milk samples were vortexed vigorously and serially diluted (1/10) in sterile phosphate-buffered saline (PBS, WELGENE, Gyeongsan, Republic of Korea). Samples were then spread onto MRS agar plates (BD Difco, Sparks, MD, USA) and incubated under anaerobic conditions for 48 h at 37 °C. To obtain pure isolates, 27 cultured colonies were picked at random and streaked onto fresh MRS agar plates. These LAB strains were maintained at −80 °C as frozen stocks in MRS broth medium (Cat. 288210, BD Difco, Sparks, MD, USA) containing 20% (*v*/*v*) glycerol. Twenty-seven strains were incubated in MRS broth at 37 °C for 24 h and then harvested by centrifugation for 20 min at 4000× *g*. Cell pellets were washed and resuspended in PBS, and then used for in vitro assays. Fresh cultured LAB strains were mass-cultured and freeze-dried for use in in vivo assays. After freeze-drying, the number of viable cells was counted. For the in vivo experiments, mice received 1 × 10^9^ CFU/day.

### 2.2. Culture of HepG2 Cells

HepG2 human hepatoma cells were purchased from the American Type Culture Collection (Manassas, VA, USA) and cultured at 37 °C/5% CO_2_ in Minimum Essential Medium (MEM; Cat. LM001-01, WELGENE, Gyeongsan, Republic of Korea) containing 10% fecal bovine serum (FBS; Cat. 10082147, Gibco, Waltham, MA, USA) and 1% penicillin–streptomycin (P/S; Cat. 15140122, Gibco, Waltham, MA, USA). We used 10 passages of HepG2 cells.

### 2.3. Treatments and Cell Viability Assays

To determine the appropriate concentration of PA solution for use in the experiments and to analyze cell viability, assays were conducted using a Cell-Counting Kit-8 assay (CCK-8; Cat. CK04, Dojindo Molecular Technologies, Inc., Kumamoto, Japan). Cells were cultured in 96-well plates at a density of 1 × 10^4^ cells/well. At 80% confluence, the medium was replaced with serum-free medium to instill starvation conditions. The 100 mM PA (Cat. 57-0-3, Sigma Aldrich, St. Louis, MO, USA) stock was prepared by dissolving the compound in 99% ethanol. The resulting PA solution (5 mM) was prepared in serum-free MEM containing 5% bovine serum albumin (BSA; Sigma Aldrich, St. Louis, MO, USA) [26,27]. PA solution was diluted to a final concentration of 0.5, 0.75, 1, or 2 mM and treated for 24 h. LAB strains were prepared in sterilized PBS. HepG2 cells were treated with each LAB strain (10^4^, 10^5^, 10^6^, 10^7^ CFU/well) for 24 h. Finally, CCK-8 solution was added to each well and the plates were incubated for 3 h at 37 °C in a CO_2_ incubator. The number of viable cells was estimated by measuring fluorescence at 450 nm (BioTek^®^ Synergy HT, Winooski, VT, USA).

### 2.4. Screening by Oil-Red-O Staining 

Oil-red-O staining was conducted to evaluate total lipid accumulation in PA-induced HepG2 cells [26,27]. HepG2 cells were seeded into 12-well plates at a density of 1 × 10^5^ cells/well and acclimatized for 24 h. Next, each LAB strain (10^7^ CFU/mL in fresh free antibiotic MEM containing 0.75 mM PA) was added. Wells treated with PA solution alone were used as negative controls. After 24 h, cells were washed twice with PBS and fixed at 4 °C for 5 min with 10% formalin solution (Cat. HT501128, Sigma Aldrich, St. Louis, MO, USA). The formalin solution was then removed and cells were fixed for 1 h in 10% formalin solution. The fixed cells were then washed twice with distilled water and once with 60% isopropanol, and then dried for 5 min. After treatment with Oil-red-O solution, the dried cells were incubated for 10 min in a dark room. The Oil-red-O stock was prepared by dissolving 0.5 g of Oil-red-O (Cat. O0625, Sigma Aldrich, St. Louis, MO, USA) in 100 mL of isopropanol. The Oil-red-O stock was then mixed with isopropanol at a ratio of 6:4 for use in the experiments. The stained cells were washed twice with distilled water and dissolved in 100% isopropanol, and absorbance was measured at 520 nm.

### 2.5. Probiotic Characterization of the LAB Strains

The hemolytic activity of the three LAB strains was tested in accordance with American Society for Microbiology guidelines. Briefly, cultured *Lactobacillus platarum* LP158, *Lactobacillus helveticus* HY7804, and *Lacticaseibacillus paracasei* LPC226 were streaked on blood agar plates supplemented with 5–7% sheep blood (Cat. MB-B1005-P50, KisanBio, Seoul, Republic of Korea). After incubating the strains at 37 °C for 48 h, hemolytic activity was confirmed. 

Antibiotic resistance tests were conducted using MIC Test Strips (Cat. 921360, Liofilchem, Via Scozia, Italy) in accordance with European Food Safety Authority (EFSA) guidelines. Nine antibiotics were tested: *Ampicillin*, *Vancomycin*, *Gentamycin*, *Kanamycin*, *Streptomycin*, *Erythromycin*, *Clindamycin*, *Tetracyclin*, *and Chloramphenicol*. LAB were plated on MRS agar plates. The inoculated strains were left at room temperature for 10–20 min to permeate into the plate. The MIC Test Strips were placed on the plates and incubated at 37 °C for 46–48 h. The lowest part of the strip on which LAB did not grow was noted as the MIC.

### 2.6. Animal Experiments

Male C57BL/6 mice (6 weeks old) were purchased from Duyeol Biotech (Gyeonggi, Republic of Korea) and housed under constant humidity (55 ± 10%) and temperature (22 ± 1 °C) conditions with a 12 h light/dark cycle. After 1 week of acclimatization, mice were assigned randomly into one of six groups, each containing seven mice: normal (Normal, AIN-93G diet), NAFLD-induced diet (NID), NAFLD with metformin (MFN, 250 mg/kg/day; positive control), NAFLD with *L. plantarum* LP158 (LP158, 10^9^ CFU/kg/day), NAFLD with *L. helveticus* HY7804 (HY7804, 10^9^ CFU/kg/day), and NAFLD with *L. paracasei* LPC226 (LPC226, 10^9^ CFU/kg/day). The NID contained 40 kcal% fat (palm oil), 20 kcal% fructose, and 2% cholesterol (D09100310, Reserch Diet, New Brunswick, NJ, USA). The ingredient composition of NID is shown Appendix A. MFN and probiotics were dissolved in 100 μL of saline and orally administered to mice for 8 weeks. Normal and NID mice received 100 μL saline. Body weight and food intake were measured every week. After 8 weeks, mice were sacrificed, and samples of blood, liver, and epididymal fat were extracted. Collected blood was allowed to stand at room temperature prior to centrifugation at 3000× *g* for 20 min to separate the serum; then, serum samples were used for blood biochemistry analysis. Extracted tissue samples were rinsed with sterilized PBS and weighed prior to storage at −80 °C. These tissues were used for histological and gene expression analyses. All animal studies were approved by the Institutional Animal Care and Committee of the hy. Co., Ltd. R&D Center, Seoul, Republic of Korea (AEC-2022-00003-Y). A flow chart of animal experiments is shown in Figure 1.

### 2.7. Blood Biochemistry Analysis

Serum samples were analyzed at T&P Bio (Gyeonggi, Republic of Korea). Serum levels of aspartate transaminase (AST), alanine transaminase (ALT), gamma glutamyl transferase (γ-GTP), alkaline phosphatase (ALP), glucose (GLU), total cholesterol (T-CHOL), triglyceride (TG), high-density lipoprotein (HDL), and low-density lipoprotein (LDL) were measured.

### 2.8. Histological Analysis

Liver tissues were collected, fixed in 10% (*v*/*v*) formalin solution, and embedded in paraffin. Sections were cut and mounted on slides prior to staining with hematoxylin and eosin (H&E). Images of the liver tissue were taken under a Zeiss Axiovert 200M microscope (Carl Zeiss AG, Thornwood, NY, USA). The degree of hepatic steatosis was scored from 1 to 4 (1, <5%; 2, 5–33%; 3, 33–66%; and 4, >66%). Histopathological analyses, including H&E staining procedures and steatosis scoring, were performed by DooYeol Biotech (Seoul, Republic of Korea).

### 2.9. RNA Extraction, cDNA Synthesis, and Gene Expression Analysis

Total RNA was extracted from HepG2 cell and mice tissues using the Easy-Spin Total RNA Extraction Kit (Cat. 17721, iNtRON Biotechnology, Seoul, Republic of Korea), and cDNA was synthesized using an Omniscript Reverse Transcription Kit (Cat. 205110, Qiagen, Hilden, Germany). The cDNA was analyzed with a QuantStudio 6-Flex Real-time PCR System (Applied Biosystems, Foster City, CA, USA) using the TaqMan^TM^ Gene Expression Master Mix (Cat. 4369016, Applied Biosystems, Waltham, MA, USA) and TaqMan Gene Expression Assays (Applied Biosystems). Table 1 lists the target genes and assay catalog numbers. Expression of all genes was normalized to that of GAPDH (HepG2 cell; Hs99999905_m1, Mice; Mm_99999915_g1).

### 2.10. Statistical Analysis

All data are presented as the mean ± standard deviation (SD) and were analyzed with one-way ANOVA followed by Tukey’s post hoc test. All analyses were conducted using GraphPad Prism 6.0 software (GraphPad Software, San Diego, CA, USA). *p* values *<* 0.05 were considered significant.

## 3. Results

### 3.1. Screening of LAB Strains Isolated from Fresh Milk 

In this study, we isolated a total of 27 LAB strains from raw milk: *Lactiplantibacillus plantarum*, 16 strains; *Lacticaseibacillus paracasei*, 4 strains; *Limosilactobacillus reuteri*, 2 strains; *Ligilactobacillus salivarius*, 2 strains; *Lactobacillus helveticus*, 1 strain; *Limosilactobacillus fermentum*, 1 strain; *Lacticaseibacillus rhamnosus*, 1 strain (Appendix A). LAB strains were investigated to determine their inhibitory effect on lipid accumulation by HepG2 cells. This cell line is generally used as an in vitro hepatoblastoma model for exploring hepatic metabolism and liver steatosis. As shown Appendix A, the final concentration of PA was 0.75 mM, which is the minimum concentration that did not affect cell viability or induce lipid accumulation. Figure 2 shows that lipid accumulation by PA-treated cells was significantly higher (137.40 ± 10.57%) than that by control cells (*p* < 0.05). In contrast, all 27 LAB strains reduced lipid accumulation by PA-induced HepG2 cells. In particular, three strains (LP158, HY7804, and LPC226) reduced accumulation significantly compared with that in PA-treated cells (96.75 ± 0.81%, 92.68 ± 6.50%, and 85.37 ± 1.63%, respectively; LP158, *p* < 0.01; HY7804 and LPC226, *p* < 0.001). Therefore, we used these three probiotics in further experiments.

### 3.2. Probiotic Characterization of LAB Strains

To determine whether LAB are safe for use as probiotics, we first investigated their hemolytic activity using a blood agar test. The results showed that none of the three selected probiotic strains had hemolytic activity (Appendix A). In addition, the antibiotic resistance of the three strains most effective at reducing lipid accumulation in HepG2 cells was measured. The MIC cut-off value recommended by the EFSA was applied. Table 2 shows the MIC values of LP158, HY7804, and LPC226; all three strains were susceptible to all antibiotics tested. Lastly, we confirmed that HepG2 cells exposed to LP158, HY7804, and LPC226 at 10^4^, 10^5^, 10^6^, and 10^7^ CFU/well showed >85% viability (Appendix A).

### 3.3. Effect of Probiotics on Gene Expression by PA-Induced HepG2 Cells

Next, we used the PA-induced HepG2 cell model to examine the effects of probiotic strains on expression of genes related to lipogenesis. As shown in Figure 3A–D, expression of *FASN*, *ACACA*, *SREBP-1c*, and *PPARγ* by negative control cells treated with PA alone significantly increased compared with the untreated group. Expression of *FASN*, *ACACA*, *SREBP-1c*, and *PPARγ* was significantly lower in HY7804 treated cells than in PA-only treated cells (*p* < 0.001). LP158 and LPC226 treatment also reduced expression of *FASN*, *ACACA*, and *SREBP-1c* (*p* < 0.001). Expression of *PPARγ* decreased in the LP158 treated group, though the reduction was not significant. In contrast, expression of *PPARγ* was not decreased in the LPC226 treated group.

Next, we examined expression of β-oxidation-related genes *ACOX1* and *CPT1A* (Figure 3E,F). Expression of mRNA encoding *ACOX1* and *CPT1A* decreased by 0.65-fold (*p* < 0.001) and 0.93-fold in the PA-only group compared with the untreated group. Expression of *ACOX1* significantly increased in all probiotic strains compared with the PA-only group (*p* < 0.001). Expression of mRNA encoding *CPT1A* was only upregulated in the HY7804 group compared with the PA-only group (*p* < 0.001).

Figure 3 shows that the efficacy of HY7804 was greater than that of LP158 with respect to its effect on expression of all genes tested, and was significantly better than LPC226 for all genes except *FASN* and *ACACA*. Therefore, HY7804 downregulates lipogenesis-related gene expression and upregulates expression of β-oxidation genes more efficiently than LP158 and LPC226.

### 3.4. Animal Experiments

#### 3.4.1. Effects of Probiotics on Body Weight and Food Efficiency Ratio in Mice Fed an NAFLD-Induced Diet

Next, we used an NAFLD-diet-induced mouse model to assess the effects of probiotic strains LP158, HY7804, and LPC226 on NAFLD. As a positive control, metformin (MFN) was used for these experiments. MFN is known to reduce not only glucose production but also fat synthesis and accumulation in the liver [28,29].

Mice in the NID group showed a weight gain of 81% compared with the normal group (*p* < 0.001). The MFN group lost weight at a level similar to that of the normal group. Body weight gain by mice treated with LP158, HY7804, and LPC226 was 39%, 69%, and 40% lower, respectively, than that of mice in the NID group (Figure 4A,B).

Figure 4C shows that the food efficiency ratio (FER) of the NID group was 78.4% higher than that of the normal diet group (*p* < 0.001), whereas the FER of the MFN, LP158, HY7804, and LPC226 groups was 19.4%, 53.7%, 23.8%, and 48.3% higher, respectively, than that of the normal group. The FER in all treated groups was significantly lower than that in the NID group (MFN and HY7804, *p* < 0.001; LP158, *p* < 0.05; LPC226, *p* < 0.005).

#### 3.4.2. Effects of Probiotics on Liver Tissue Morphology and Liver and Epididymal Fat Mass in NAFLD-Induced-Diet Mice

Figure 3D shows that the mass of liver tissue from mice in the NID group was significantly greater (2.96 ± 0.39 g) than that from mice in the normal group (1.37 ± 0.07 g, *p* < 0.001). The weight of liver tissue from mice in the MFN (1.80 ± 0.19 g), LP158 (1.95 ± 0.15 g), HY7804 (1.87 ± 0.14 g), and LPC226 (2.15 ± 0.34 g) groups was significantly lower than that from the NID group (*p* < 0.001). The liver tissue/body weight (L/B) ratio in the NID group was significantly higher than that in the normal group (*p* < 0.001). Ingestion of MFN, LP158, HY7804, and LPC226 significantly reduced the L/B ratio; however, although LPC226 also reduced the L/B ratio, the result was not significant (Figure 4E).

We also measured epididymal fat tissue mass (Figure 4F). Epididymal fat mass in the NID group was higher (2.96 ± 0.39 g) than that in the normal group (1.37 ± 0.07 g; *p* < 0.001). Epididymal fat mass in the MFN and HY7804 groups (1.76 ± 0.31 g and 2.02 ± 0.14 g, respectively) was significantly lower than that in the NID group (*p* < 0.001); however, although LP158 and LPC226 reduced epididymal fat mass, this reduction was not significant. The results for the epididymal fat/body weight (EF/B) ratio were similar (Figure 4G).

#### 3.4.3. Effects of Probiotics on Serum Biochemistry

The levels of liver function markers ALT, AST, ALP, and γ-GPT increased significantly in the NID group compared with the normal diet group (*p* < 0.001) (Figure 5A–D). Levels of all markers in the MFN group were significantly lower than those in the NID group. Also, LP158, HY7804, and LPC226 led to a significant reduction in levels of these markers when compared with the NID group. Thus, all three probiotic groups showed improved liver function (similar to the MFN group), with no significant difference between the three.

GLU levels increased slightly in the NID group compared with the normal group, whereas they were lower in the MFN, LP158, HY7804, and LPC226 groups than in the NID group (Figure 5E). However, GLU level results showed no significant differences between groups. Serum T-CHO and LDL levels in the NID group were higher than those in the normal group (*p* < 0.001). Intake of MFN, LP158, HY7804, or LPC226 improved T-CHO and LDL levels compared with the NID group (*p* < 0.001, T-CHO, LP158; *p* < 0.01) (Figure 5F,G). HDL levels in the NID group were higher than those in the normal group, whereas those in the MFN, HY7804, and LPC226 were higher than those in the NID group, although the difference was not significant (Figure 5H). TG levels in the NID group were 198.57 ± 20.00 mg/dL compared with 161.43 ± 13.06 mg/dL in the normal group (*p* < 0.001). TG levels in all bacteria-treated groups were slightly lower than those in the NID group, although the difference was not significant (Figure 5I).

#### 3.4.4. Effects of Probiotics on Liver Tissue Morphology and Histological Analysis of Liver Tissues from NAFLD Mice

Liver tissue morphology is presented in Figure 6A. Tissue from the NID group was visually whiter and larger than that from the normal group. Visual inspection suggested that all three probiotics improved the size and color of the liver tissue, which resembled that from the positive control (MFN) group.

To analyze differences in hepatic histology, we examined sections of liver tissue samples after H&E staining. As shown in Figure 6B, severe hepatic steatosis was observed in the NID group, whereas it was alleviated in the LAB-treated groups. Histopathological analysis revealed that the steatosis grade was significantly lower in all LAB groups than in the NID group. In particular, the HY7804 group showed the greatest reduction in steatosis; there was no statistically significant difference between the other LAB-treated groups (Figure 6C). In the MFN group, hepatic steatosis tended to be less severe than in the NID group, but the difference was not significant.

#### 3.4.5. Effects of Probiotics on Gene Expression by NAFLD Mice

Finally, we examined the effects of probiotics on expression of hepatic genes, focusing on those related to lipid synthesis, β-oxidation, fibrosis, and inflammation (Figure 7). First, we examined the expression of genes related to lipid synthesis (*Fasn*, *Pparγ*, *Srebp-1*, *Acaca*, and *Scd*) (Figure 7A). The NID group showed higher expression of *Fasn*, *Pparγ*, *Srebp-1*, and *Scd1* than the normal group (*p* < 0.001). Expression of *ACACA* also increased in the NID group, but the increase was not significant. Expression of mRNA encoding *Fasn* was suppressed by MFN, LP158, HY7804, and LPC226 compared with that in the NID group (*p* < 0.001). Expression of *Pparγ*, *Srebp-1*, *Acaca*, and *Scd1* fell significantly in all treated groups, except for LPC226. In particular, HY7804 decreased expression of lipid-synthesis-related genes to a level almost similar to that of MFN. Expression of mRNA encoding *Acaca* (0.71-fold, *p* < 0.001) and *Scd1* (1.84-fold, *p* < 0.05) fell significantly in the HY7804 group compared with the LPC226 group (*Acaca*; 1.13-fold, *Scd1*; 2.98-fold). Furthermore, expression of *Srebp-1* in the HY7804 group (0.87-fold) was significantly lower than that in the LP158 group (1.31-fold, *p* < 0.05) and LP226 group (1.43-fold, *p* < 0.001).

In addition, we examined expression of fatty acid β-oxidation-related genes (*Acox1*, *Cpt1a*, and *Pparα*) (Figure 7B). Expression of *Acox1* and *Cpt1a* in the NID group was significantly lower than in the normal group. Intake of LP158 and LPC226 upregulated expression of *Acox1* and *Cpt1a*, but at a lower level than that observed in the HY7804 group. Expression was highest in the HY7804 group; indeed, it was even higher than that in the positive control group treated with MFN. While expression of *Pparα* fell slightly, it was similar to that in the normal diet group. Expression of *Pparα* in the LP158 and HY7804 groups was significantly higher than that in the NID group, and both were higher than that in the MFN diet group.

As shown in Figure 7C, we also evaluated expression of mRNA level-encoding inflammation markers (a macrophage marker (*Cd68*), pro-inflammatory cytokines (*Tnfα* and *Il-1β*), and a chemokine (*Ccl2*)). Expression of these genes in the NID group was higher than that in the normal group. Thus, the NID significantly increased expression of inflammation-related genes (*Cd68*, *p * < 0.01; *Tnf*, *p* < 0.001; *Ccl2*, *p* < 0.001) compared with that in the normal group. Expression of inflammation-related genes in the LAB-treated groups was lower than that in the NAFLD diet group. Among these, HY7804 showed the greatest inhibitory effect (similar to MFN).

Finally, we examined expression of fibrosis-related genes *Col1a1*, *Col3a1*, and *Tgfβ1* (Figure 7D). All three factors were upregulated in the NID group compared with the normal group (*Col1a1* and *Col3a1; p* < 0.001, *Tgfβ1*; n.s.). Probiotics reduced the hepatic abundance of mRNAs encoding *Col1a1*, *Col3a1*, and *Tgfβ1*; however, there was no significant difference between LP158, HY7804, and LPC226, and only HY7804 decreased expression of these fibrosis-related genes significantly when compared with the NID group.

## 4. Discussion

NAFLD, the most common cause of liver dysfunction (including liver steatosis and liver cancer), has a prevalence of up to 20% worldwide [30]. Unlike fatty liver, which is characterized simply by fat accumulation, NAFLD is characterized by inflammation and fibrosis [4,5,31]; however, no treatment has been approved by the US Food and Drug Administration (FDA) or the European Medicines Agency (EMA). At present, NAFLD is treated using drugs designed to treat diabetes, obesity, and hyperlipidemia [32,33]. Recent studies suggest that probiotics may prevent and/or improve NAFLD [4,5,34].

*Lactiplantibacillus plantarum* LP158, *Lactobacillus helveticus* HY7804, and *Lacticaseibacillus paracasei* LPC226 were isolated from fresh milk obtained from domestic animals in Korea. These three novel strains were selected from among twenty-seven LAB strains, and all three showed significant anti-fat-accumulation effects in HepG2 cells. We next assessed whether LP158, HY7804, and LPC226 were toxic to HepG2 cells. Hemolytic activity in humans is potentially dangerous [35], and antibiotic resistance is a public health concern worldwide. Previous studies showed that antibiotic resistance can transfer between pathogens and probiotic bacteria, either directly or indirectly, via the human intestinal flora [36,37,38]. Here, we found that LP158, HY7804, and LPC226 showed no α-, β-, or γ-hemolytic effects, nor were they resistant to any of the nine antibiotics tested.

We also examined the effects of the three LAB strains on expression of lipogenesis- and β-oxidation-related genes in PA-induced HepG2 cells. We found that expression of mRNA encoding lipogenesis-related genes *FANS*, *ACACA*, *SREBP-1c*, and *PPARγ* decreased upon exposure to LP158, HY7804, and LPC226. Studies show that these genes are associated with NAFLD and contribute to increased fatty acid synthesis [39,40]. FASN catalyzes biosynthesis of fatty acids from acetyl-CoA and malonyl-CoA. ACACA plays a role in production of malonyl-CoA for fatty acid synthesis, and it suppresses mitochondrial fatty acid oxidation by blocking expression of carnitine palmitoyltransferase 1A (CPT1A) [17]. Transcription factor SREBP-1c regulates expression of genes related to lipogenesis and the glycolysis pathway in the liver. SREBP-1c affects TG synthesis in the liver and adipose tissues [16]. PPARγ stimulates lipid uptake and adipogenesis, and it induces storage fatty acids in the liver [41]. Fatty acids commonly move from the cytosol to the mitochondria, where they undergo β-oxidation [42]. CPT1A promotes passage through the mitochondrial membrane, allowing activated fatty acids in the cytosol to move into the mitochondria [43]. ACOX1 is mainly involved in the fatty acid β-oxidation pathway [42]. Here, we found that treatment of PA-induced HepG2 cells with probiotic strains increased expression of β-oxidation-related genes such as *CPT1A* and *ACOX1*. In other words, the results suggest that the three probiotic strains reduce hepatic fatty acid synthesis and improve β-oxidation in the liver.

Next, we examined the effect of the three selected probiotics in a mouse model of NAFLD. Metformin was used as a positive control for these animal experiments. Metformin is used to treat type 2 diabetes, but also alleviates hepatic lipogenesis in the NAFLD mouse model [29].

After 8 weeks, NID mice showed a significant increased body weight, as well as mass gain in liver tissues. Treatment with either LP158 or LPC226 alone reduced epididymal fat mass. In particular, treatment with HY7804 reduced it to levels similar to those in the MFN group. Thus, mice in the LP158, HY7804, and LPC226 groups showed lower weight gain, as well as lower liver and epididymal fat mass, than mice in the NID group.

Serum ALT, AST, ALP, and γ-GPT levels were significantly lower in the LP158, HY7804, and LPC226 groups than in the NID group; indeed, levels in all three LAB-treated groups were similar to those in the MFN group. These markers may be useful as measurement factors for NAFLD. Increased ALT and AST levels are found in patients with many types of NAFLD, ranging from simple steatosis to NASH [44]. Elevated ALP levels in the blood are most commonly observed in those with liver cancer, cirrhosis, or hepatitis [45]. Previous studies showed that γ-GPT is strongly associated with liver fat severity, and there is a significant relationship between elevated γ-GPT and increased incidence of NAFLD [44,45,46]. Therefore, LP158, HY7804, and LPC226 may improve NAFLD by regulating hepatic functional markers in serum. Levels of serum total cholesterol and triglycerides, which correlate with the incidence of NAFLD, increased in the NAFLD model mice [47]. The three selected probiotics decreased glucose and triglyceride levels, and increased HDL levels, in blood. NAFLD causes overproduction of glucose, leading to abnormal blood sugar levels and insulin resistance [48]. In obese or overweight individuals, alterations in hepatic fatty acid metabolism promote accumulation of hepatic triglycerides [49,50]. These data suggest that treatment with the three selected probiotics improves symptoms of NAFLD by downregulating lipid and glucose levels in the blood. In addition, H&E staining of liver tissues demonstrated that intake of the selected probiotics by NAFLD mice reduced hepatic fat accumulation.

Increased expression of lipogenic genes in NAFLD-induced mice confirmed that liver fatty acid synthesis occurs actively in this model. Pparγ plays a role in the synthesis of TG and modulates fat accumulation by regulating production of insulin-sensitive adipokines such as adiponectin [51,52]. Scd1 is the rate-limiting enzyme for de novo lipogenesis, and it plays a role in increasing production and accumulation of triglycerides in the liver [53]. Expression of mRNA encoding these genes fell after treatment with the selected probiotics. These results suggest that LP158, HY7804, and LPC226 increase expression of mRNA encoding β-oxidation-related genes *Acox1* and *Cpt1a*. Regarding β-oxidation, Pparα is also a major factor and fatty acid sensor that modulates transcription of genes encoding the rate-limiting enzymes for hepatic lipid oxidation [54]. A previous study shows that lack of Pparα in hepatocytes may spontaneously trigger liver steatosis, suggesting that hepatic steatosis may worsen in those with NASH [55]. The data in the present study showed that expression of *Pparα* mRNA was elevated by the LAB bacteria in the order HY7804 > LP158 > LPC226, a pattern similar to the degree of decrease in steatosis grade.

Expression of mRNA encoding the macrophage marker *Cd68*, pro-inflammatory cytokines *Il-1β* and *Tnfα*, and the chemokine *Ccl2* in mouse liver was lower in the LP158, HY7804, and LPC226 groups than in the NAFLD group. Inflammatory factors make a significant contribution to development and progression of NAFLD [56]. Hepatic lipid accumulation leads to progression of liver inflammation and induces production of cytokines and chemokines [18,57]. Probiotic-treated groups showed reduced levels of mRNAs associated with lipogenesis and steatosis, and also lower inflammation gene expression than NID mice. Liver fibrosis in NAFLD is caused by liver inflammation, and major markers of fibrosis are *Col1a1*, *Col3a1*, and *Tgfβ1* [20,58]. *Col1a1* and *Col3al* encode COLⅠ and COL Ⅲ, which are major components of fibrotic tissue in the liver [58]. Tgfβ1 plays a role in collagen production, and it strongly promotes collagen accumulation and fibrotic tissue responses [59]. Expression of fibrosis-related gene expression fell in mice treated with the selected probiotics. These data show that LP158, HY7804, and LPC 226 inhibit hepatic fibrosis in the mouse liver.

## 5. Conclusions

The in vitro and in vivo experiments performed herein show that, in the liver, LAB strains LP158, HY7804, and LPC226 not only suppress steatosis, inflammation, and fibrosis but also activate β-oxidation. Among them, *Lactobacillus helveticus* HY7804 was the best at alleviating NAFLD symptoms. Thus, *Lactobacillus helveticus* HY7804 isolated from raw milk is a potential probiotic strain that may prevent NAFLD. Thus, milk or dairy products containing HY7804 may have health benefits with respect to treatment/prevention of NAFLD.

## Figures and Tables

**Figure 1 microorganisms-11-02466-f001:**
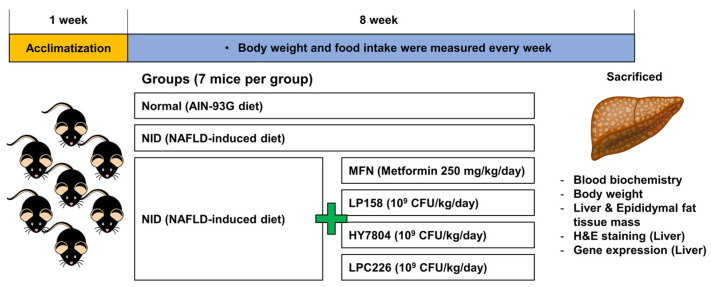
Flow chart showing the animal experiments.

**Figure 2 microorganisms-11-02466-f002:**
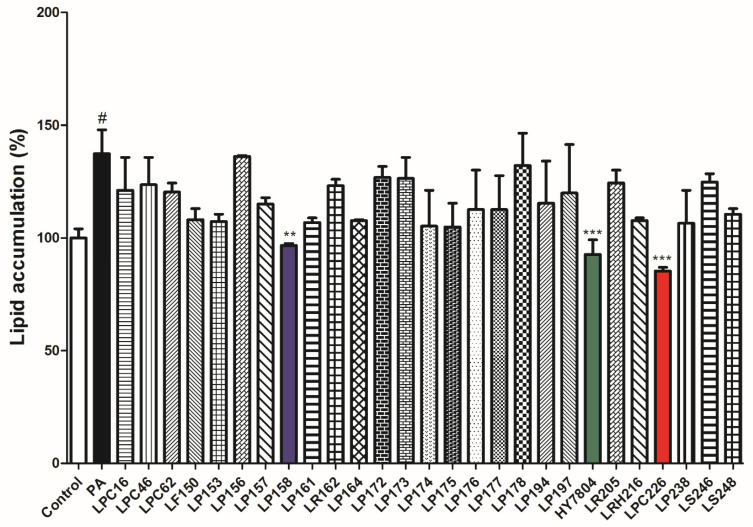
Screening of 27 LAB strains for their ability to reduce lipid accumulation in PA-induced HepG2 cells. Results are presented as the mean ± SD. Significant differences are indicated as ^#^
*p* < 0.05 compared with the untreated group; and ** *p* < 0.01 and *** *p* < 0.001 compared with the PA group. PA, palmitic acid treatment group.

**Figure 3 microorganisms-11-02466-f003:**
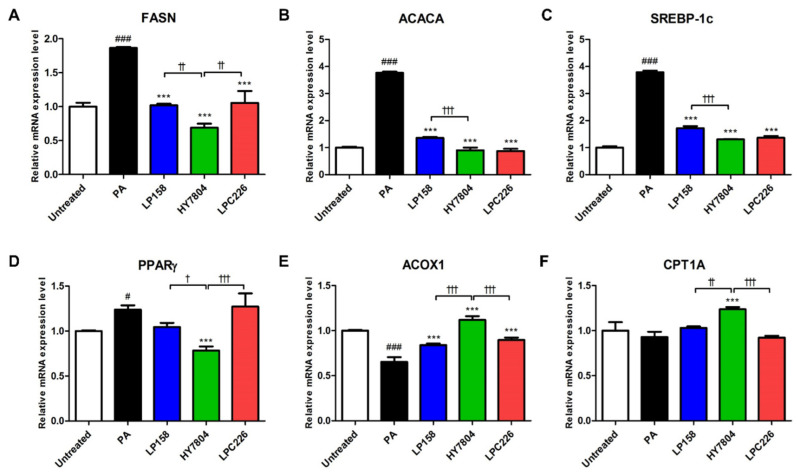
Effect of probiotics on gene expression by PA-induced HepG2 cells. (**A**) FASN, (**B**) ACACA, (**C**) SREBP-1c, (**D**) PPARγ, (**E**) ACOX1, and (**F**) CPT1A. Results are presented as the mean ± SD. Significant differences are indicated as ^#^
*p* < 0.05, and ^###^
*p* < 0.001 compared with the Untreated group; *** *p* < 0.001 compared with the PA group; and ^†^
*p* < 0.05, ^††^
*p* < 0.01, and ^†††^
*p* < 0.001 between probiotic groups. PA, palmitic acid treatment group; LP158, *Lactiplantibacillus plantarum* LP158; HY7804, *Lactobacillus helveticus* HY7804; LPC226, *Lacticaseibacillus paracasei* LPC226.

**Figure 4 microorganisms-11-02466-f004:**
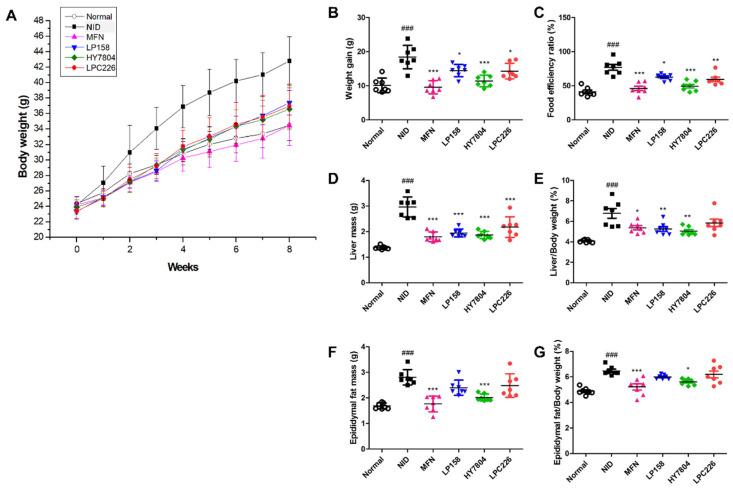
Effect of probiotics on NAFLD-diet-induced mice. (**A**) Changes in body weight, (**B**) weight gain, (**C**) food efficiency ratio, (**D**) liver mass, (**E**) liver/body weight ratio, (**F**) epididymal fat mass, and (**G**) epididymal fat/body weight ratio are shown. Results are presented as the mean ± SD. Significant differences are indicated as ^###^
*p* < 0.001 compared with the normal group, and * *p* < 0.05, ** *p* < 0.01, and *** *p* < 0.001 compared with the NAFLD diet group. NID, NAFLD-induced diet; MFN, metformin; LP158, *Lactiplantibacillus plantarum* LP158; HY7804, *Lactobacillus helveticus* HY7804; LPC226, *Lacticaseibacillus paracasei* LPC226.

**Figure 5 microorganisms-11-02466-f005:**
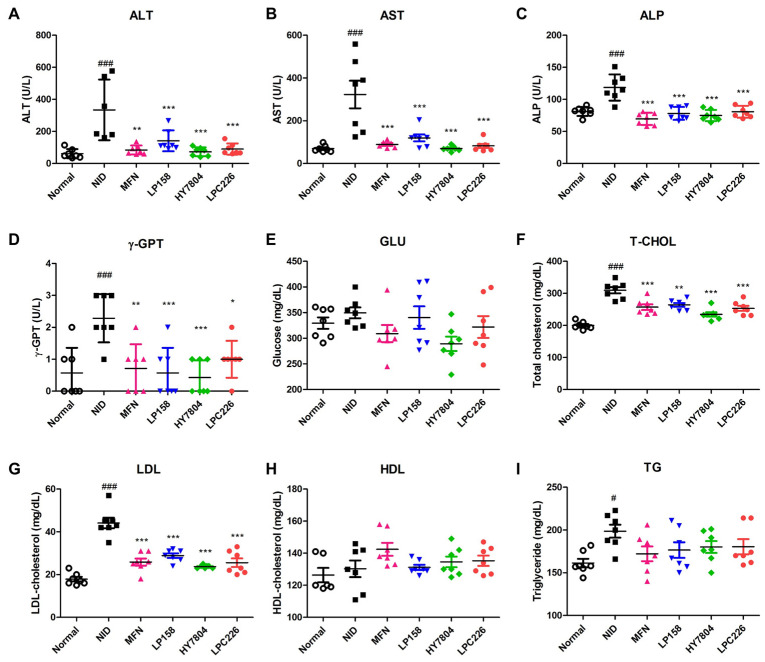
Effect of probiotics on serum biochemistry in NAFLD-diet-induced mice. (**A**) ALT, (**B**) AST, (**C**) ALP, (**D**) γ-GPT, (**E**) GLU, (**F**) T-CHO, (**G**) LDL, (**H**) HDL, and (**I**) TG levels are shown. Results are presented as the mean ± SD. Significant differences are indicated as ^#^
*p* < 0.05 and ^###^
*p* < 0.001 compared with the normal group, and * *p* < 0.05, ** *p* < 0.01, and *** *p* < 0.001 compared with the NAFLD diet group. NID, NAFLD-induced diet; MFN, metformin; LP158, *Lactiplantibacillus plantarum* LP158; HY7804, *Lactobacillus helveticus* HY7804; LPC226, *Lacticaseibacillus paracasei* LPC226; ALT, alanine transferase; AST, aspartate transferase; ALP, alkaline phosphatase; γ-GTP, gamma glutamyl transferase; GLU, glucose; T-CHO, total-cholesterol; LDL, low-density lipoprotein cholesterol; HDL, high-density lipoprotein cholesterol; TG, triglyceride.

**Figure 6 microorganisms-11-02466-f006:**
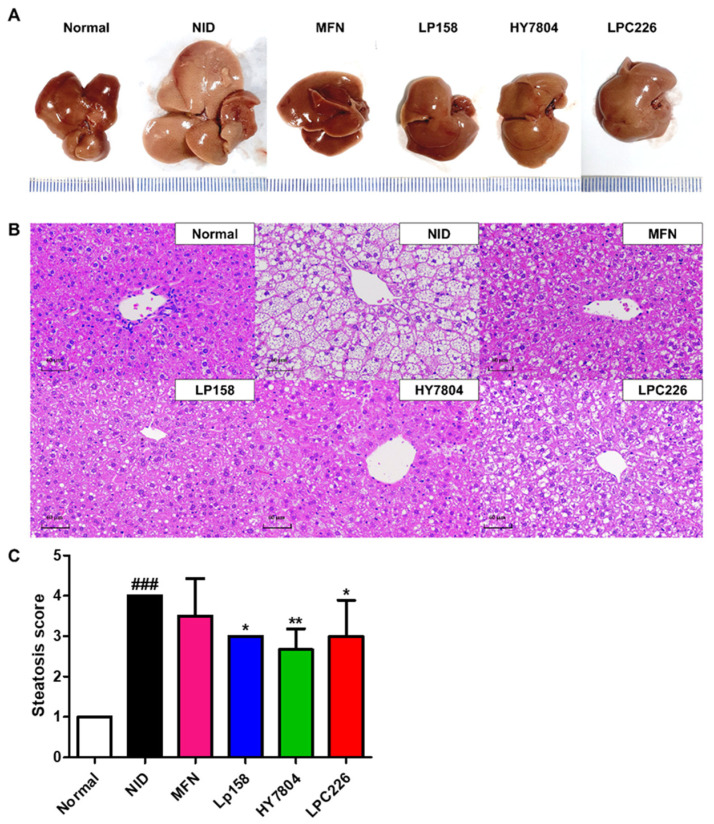
Effect of probiotics on liver tissue morphology, and histological analysis of H&E-stained tissue sections. (**A**) Liver tissue morphology. (**B**) Histological changes (H&E-stained, 100 × magnification). (**C**) Steatosis score in liver tissues from NAFLD mice. Results are presented as the mean ± SD. Significant differences are indicated as ^###^
*p* < 0.001 compared with the normal group. * *p* < 0.05 and ** *p* < 0.01 compared with the NAFLD diet group. NID, NAFLD-induced diet; MFN, metformin; LP158, *Lactiplantibacillus plantarum* LP158; HY7804, *Lactobacillus helveticus* HY7804; LPC226, *Lacticaseibacillus paracasei* LPC226.

**Figure 7 microorganisms-11-02466-f007:**
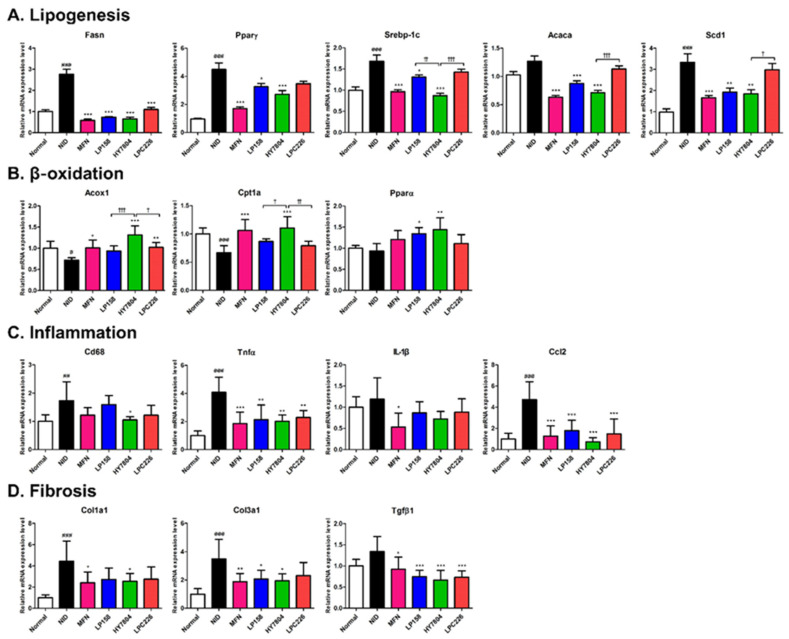
Effect of probiotics on expression of genes in the liver. Expression of genes related to (**A**) lipogenesis, (**B**) β-oxidation, (**C**) inflammation, and (**D**) fibrosis is shown. Results are presented as the mean ± SD. Significant differences are indicated as ^#^
*p* < 0.05, ^##^
*p* < 0.01, and ^###^
*p* < 0.001 compared with the normal group; * *p* < 0.05, ** *p* < 0.01, and *** *p* < 0.001 compared with the NAFLD diet group; and ^†^
*p* < 0.05, ^††^
*p* < 0.01, and ^†††^
*p* < 0.001 between the probiotic groups. NID, NAFLD-induced diet; MFN, metformin; LP158, *Lactiplantibacillus plantarum* LP158; HY7804, *Lactobacillus helveticus* HY7804; LPC226, *Lacticaseibacillus paracasei* LPC226.

**Table 1 microorganisms-11-02466-t001:** TaqMan probes used for generic analysis, along with their catalog numbers.

Gene	Gene Name	Catalog Number
Human		
*GAPDH*	Glyceraldehyde-3-phosphate dehydrogenase	Hs99999905_m1
*FASN*	Fatty acid synthase	Hs00188012_m1
*SREBP-1*	Sterol regulatory element-binding protein 1	Hs01088691_m1
*PPARγ*	Peroxisome proliferator-activated receptor gamma	Hs01115513_m1
*C/EBPa*	CCAAT/enhancer-binding protein alpha	Hs00269972_s1
*ACACA*	Acetyl-CoA carboxylase alpha	Hs01046047_m1
*CPT1A*	Carnitine palmitoyltransferase 1A	Hs00912671_m1
*ACOX1*	Acyl-CoA oxidase 1	Hs01074241_m1
Mouse		
*Gapdh*	Glyceraldehyde-3-phosphate dehydrogenase	Mm99999915_g1
*Fasn*	Fatty acid synthase	Mm00433237_m1
*Srebp-1*	Sterol regulatory element-binding protein 1	Mm00550338_m1
*Pparγ*	Peroxisome proliferator-activated receptor gamma	Mm00440945_m1
*C/ebpα*	CCAAT/enhancer-binding protein alpha	Mm00514283_m1
*Acaca*	Acetyl-CoA carboxylase alpha	Mm01304257_m1
*Scd1*	Steroyl-Coenzyme A desaturease 1	Mm00772290_m1
*Cpt1a*	Carnitine palmitoyltransferase 1a	Mm01231183_m1
*Acox1*	Acyl-CoA oxidase 1	Mm01246831_m1
*Col1a1*	Collagen type III alpha 1	Mm00801666_g1
*Co13a1*	Collagen type I alpha 1	Mm00802300_m1
*Tgf* *β* *1*	Transforming growth factor beta 1	Mm01178820_m1
*Cd68*	CD68 molecule	Mm03047343_m1
*Ccl2*	C-C motif chemokine ligand 2	Mm00441242_m1
*Tnf*	Tumor necrosis factor	Mm00443258_m1
*Il-1b*	Interleukin 1 beta	Mm00434228_m1
*Pparα*	Peroxisome proliferator-activated receptor alpha	Mm00440939_m1

**Table 2 microorganisms-11-02466-t002:** MIC cut-off values for the three LAB strains.

Antibiotic	EFSA ^1^	LP158	EFSA	HY7804	EFSA	LPC226
*Ampicillin*	2	0.032	1	0.125	4	0.5
*Vancomycin*	n.r. ^2^	n.r.	2	0.5	n.r.	n.r.
*Gentamycin*	16	12	16	16	32	16
*Kanamycin*	64	64	16	16	64	64
*Streptomycin*	n.r.	n.r.	16	16	64	64
*Erythromycin*	1	0.75	1	0.25	1	1
*Clindamycin*	2	0.75	1	0.75	1	0.25
*Tetracyclin*	32	8	4	2	4	2
*Chloramphenicol*	8	6	4	4	4	3

^1^ EFSA, European Food Safety Authority; ^2^ n.r., not required.

## Data Availability

The data presented in this study are available in the article and Appendix A.

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
