# Peer review of "Lactobacillus helveticus Isolated from Raw Milk Improves Liver Function, Hepatic Steatosis, and Lipid Metabolism in Non-Alcoholic Fatty Liver Disease Mouse Model"

_microorganisms, 2023, doi:10.3390/microorganisms11102466_

Round 1

Reviewer 1 Report

In this manuscript, Hyeonji Kim et al. describe a set of experiments suggesting that Lactobacillus plantarum LP158 (LP158), Lactobacillus helveticus HY7804 (HY7804), and Lactobacillus paracasei LPC226 (LPC226) isolated from raw milk alleviate nonalcoholic fatty acid disease (NAFLD) in a diet-induced C57BL/6 mouse model. It was demonstrated that HY7804, LP158 and LPC226 can ameliorate NAFLD through anti-steatosis, anti-inflammatory and anti-fibrotic effects. This is due to their ability to regulate mRNA encoding lipogenesis genes (Fasn, Acaca and Srebp-1c) and inflammatory (Tnfα and Ccl-2) and fibrosis factors and the expression of a β-oxidation gene (Acox1) in mouse liver, accompanied by a modulation of blood biochemical markers of liver damage and a reduction in steatosis in liver tissue. The article has merit and presents some interesting data.

Comments

·       Authors should provide the catalog number of all reagents and kits used in the study for reproducibility purposes.

·       Authors should describe the pass number of the cell culture in which the experiments were performed.

·       The authors suggest that HY7804, LP158 and LPC226 can ameliorate NAFLD through anti-steatosis, anti-inflammatory and anti-fibrotic effects. The authors could support this hypothesis with complementary studies on mouse liver tissue to measure the degree of fibrosis with histological techniques such as Masson's trichrome staining, or Sirius red to measure collagen deposition in the liver, as well as Oil-red-O Staining to stain lipids in the liver.

·       On line 298-300 the following statement is presented "GLU levels increased slightly in the NID group compared with the Normal group, whereas they were lower in the MFN, LP158, HY7804, and LPC226 groups than in the NID group (Figure 5E). However, the figure does not show a statistically significant difference, which should be described in the results.

·       In the description of the murine model of NAFLD-induced diet (NID), I suggest that the authors better describe the murine model, in this model from which week an increase in inflammation and fibrosis is observed, it is important to properly highlight this information because the treatments with Lactobacillus helveticus were started at the same time as the diet to induce NAFLD, therefore, it would be hypothesized that the effect of Lactobacillus helveticus would be anti-steatosis, and anti-inflammatory, while the anti-fibrotic effect should be evaluated when a fibrotic state was already established in the liver of the mice. What is the justification for administering Lactobacillus helveticus treatment at the same time as diet to induce NAFLD, is it a preventive treatment, or a reversal treatment?

Reviewer 2 Report

The writing of this research is very clear, and the experimental results support the author's hypothesis. The experimental data and results are quite impressive, as almost all tested parameters could be improved by the relevant strains. However, the deterioration of relevant indicators in just 8 weeks using the NAFLD-diet is somewhat inconsistent with the experiences of most animal models of NAFLD found in the literature. It is recommended that the author should specify the name and manufacturer of the feed used, as this information can be useful for others when designing similar experiments. I suggest the author could briefly discuss their model.

1.          In recent years, the nomenclature for certain Lactobacillus species has undergone changes. For updated species names, please refer to the most current information available at: https://isappscience.org/new-names-for-important-probiotic-lactobacillus-species/

2.          On lines 71-72, the authors mentioned, "These three species were selected from 27 LAB strains based on their ability to inhibit fat accumulation in Palmitic Acid (PA)-treated HepG2 cells." It is important to provide an explanation of why "inhibitory effects on fat accumulation in PA-treated HepG2 cells" were chosen as a screening model. Additionally, this choice should be supported by a citation from a relevant previous study. While the authors have elaborated on the role of PA-treated cells in the results section, it is advisable to include this explanation in the materials and methods section as well.

3.          In lines 99-100, the roles of Palmitic Acid (PA) analysis should be elucidated, and a citation should be included to substantiate the use of PA analysis in the study.

4.          Between lines 111 and 123, the purpose of conducting "Oil-red-O Staining analysis" should be clarified within this section.

5.          At line 128, it is important to specify the type of blood agar used for the hemolytic activity analysis. The choice of blood agar medium can significantly impact the sensitivity of hemolytic activity analysis. If the blood agar plate was prepared by mixing blood with "MRS agar," this information should be included in the manuscript.

6.          On lines 147-148, the authors mentioned, "The NID contained 40 kcal% fat (palm oil), 20 kcal% fructose, and 2% cholesterol." It is strongly recommended that the authors provide the brand name of the diets used, as this specific treatment appears to efficiently induce NAFLD in mice within just 8 weeks. Additionally, a brief discussion of their animal model is suggested.

7.          At line 255, where it is stated, "MFN reduces the synthesis of glucose and fat in the liver," please add a citation to substantiate this information.

8.          Why did the authors not conduct insulin resistance, glucose tolerance tests, and HOMA-IR analysis in their animal model? These assessments are crucial for metabolic-related studies involving animal models.

Round 2

Reviewer 1 Report

The authors have adequately addressed each of the observations and suggestions made.

Reviewer 2 Report

I am content with the authors' replies.